# Heavy Metal (Arsenic) Induced Antibiotic Resistance among Extended-Spectrum β-Lactamase (ESBL) Producing Bacteria of Nosocomial Origin

**DOI:** 10.3390/ph15111426

**Published:** 2022-11-17

**Authors:** Naveed Ahmed, Kinza Tahir, Sara Aslam, Sara Masood Cheema, Ali A. Rabaan, Safaa A. Turkistani, Mohammed Garout, Muhammad A. Halwani, Mohammed Aljeldah, Basim R. Al Shammari, Amal A. Sabour, Maha A. Alshiekheid, Saleh A. Alshamrani, Reyouf Al Azmi, Ghadeer H. Al-Absi, Shah Zeb, Chan Yean Yean

**Affiliations:** 1Department of Medical Microbiology and Parasitology, School of Medical Sciences, Universiti Sains Malaysia, Kubang Kerian 16150, Kelantan, Malaysia; 2Department of Microbiology, Faculty of Life Sciences, University of Central Punjab, Lahore 54000, Pakistan; 3Department of Medical Education, Allama Iqbal Medical College, Lahore 54000, Pakistan; 4Department of Pathology, Azra Naheed Medical College, Lahore 54000, Pakistan; 5Molecular Diagnostic Laboratory, Johns Hopkins Aramco Healthcare, Dhahran 31311, Saudi Arabia; 6College of Medicine, Alfaisal University, Riyadh 11533, Saudi Arabia; 7Department of Public Health and Nutrition, The University of Haripur, Haripur 22610, Pakistan; 8Department of Medical Laboratory Sciences, Fakeeh College for Medical Science, Jeddah 21134, Saudi Arabia; 9Department of Community Medicine and Health Care for Pilgrims, Faculty of Medicine, Umm Al-Qura University, Makkah 21955, Saudi Arabia; 10Department of Medical Microbiology, Faculty of Medicine, Al Baha University, Al Baha 4781, Saudi Arabia; 11Department of Clinical Laboratory Sciences, College of Applied Medical Sciences, University of Hafr Al Batin, Hafr Al Batin 39831, Saudi Arabia; 12Department of Botany and Microbiology, College of Science, King Saud University, Riyadh 11451, Saudi Arabia; 13Department of Clinical Laboratory Sciences, College of Applied Medical Sciences, Najran University, Najran 61441, Saudi Arabia; 14Infection Prevention and Control, Eastern Health Cluster, Dammam 32253, Saudi Arabia; 15College of Pharmacy, Department of Pharmacy Practice, Alfaisal University, Riyadh 325476, Saudi Arabia; 16Department of Microbiology, Faculty of Biomedical and Health Science, The University of Haripur, Haripur 22610, Pakistan

**Keywords:** ESBL, sodium arsenate, heavy metals, antibiotics resistance, antimicrobial resistance, double disk diffusion test

## Abstract

Antimicrobial resistance (AMR) is a leading cause of treatment failure for many infectious diseases worldwide. Improper overdosing and the misuse of antibiotics contributes significantly to the emergence of drug-resistant bacteria. The co-contamination of heavy metals and antibiotic compounds existing in the environment might also be involved in the spread of AMR. The current study was designed to test the efficacy of heavy metals (arsenic) induced AMR patterns in clinically isolated extended-spectrum β-lactamase (ESBL) producing bacteria. A total of 300 clinically isolated ESBL-producing bacteria were collected from a tertiary care hospital in Lahore, Pakistan, with the demographic characteristics of patients. After the collection of bacterial isolates, these were reinoculated on agar media for reidentification purposes. Direct antimicrobial sensitivity testing (AST) for bacterial isolates by disk diffusion methods was used to determine the AST patterns with and without heavy metal. The heavy metal was concentrated in dilutions of 1.25 g/mL. The collected bacterial isolates were isolated from wounds (*n* = 63, 21%), urine (*n* = 112, 37.3%), blood (*n* = 43, 14.3%), pus (*n* = 49, 16.3%), and aspirate (*n* = 33, 11%) samples. From the total 300 bacterial isolates, *n* = 172 were *Escherichia coli* (57.3%), 57 were *Klebsiella* spp. (19%), 32 were *Pseudomonas aeruginosa* (10.6%), 21 were *Proteus mirabilis* (7%) and 18 were *Enterobacter* spp. (6%). Most of the antibiotic drugs were found resistant to tested bacteria. Colistin and Polymyxin-B showed the highest sensitivity against all tested bacteria, but when tested with heavy metals, these antibiotics were also found to be significantly resistant. We found that heavy metals induced the resistance capability in bacterial isolates, which leads to higher AMR patterns as compared to without heavy metal tested isolates. The results of the current study explored the heavy metal as an inducer of AMR and may contribute to the formation and spread of AMR in settings that are contaminated with heavy metals.

## 1. Introduction

The rising cases of antimicrobial resistance (AMR) in bacteria are threatening the potency of antibiotics, which have revolutionized therapies, and threaten millions of lives [1]. The AMR has reached the generation of antibiotics after the emergence of the first resistance cases against penicillin and other drugs [2]. The AMR has been linked to misuse and improper use of such drugs, in addition to a shortage of novel treatment products by the biopharmaceutical sector, as a result of diminished financial remuneration and difficult compliance standards [3,4]. The Centers for Disease Control and Prevention (CDC) has identified a variety of pathogens as severe and pose alarming risks, several of which are already imposing a major interventional and economic impact on the United States (US) healthcare system, patients, and their communities [5].

Bacteria are more likely to acquire AMR as antibiotics are used more frequently. As a result, antibiotics will be ineffective when we need them in the future [6]. If we reduce the use of antibiotics, there might be a chance to reduce the prevalence of high AMR rates [4,7]. By the use of antibiotics, certain bacteria die, while resistant bacteria thrive and ultimately multiply. Antibiotic excessive use raises the prevalence of resistant bacteria [8]. The World Health Organization (WHO) survey presumed that the issue was connected with the pervasiveness and abundance of resistant microorganisms and genes in bacteria [9]. 

The extended-spectrum β-lactamase (ESBL) are the enzymes that express by genes located on the plasmids [10]. They show strong hydrolytic activity against aztreonam, cephalosporins and penicillin and play a vital role in multi-resistant (MDR) gram-negative bacteria [11]. The ESBLs are classified into three major groups such as CTX-M, SHV and TEM types [10]. *Escherichia coli* [12], *Klebsiella* spp. [12], *Pseudomonas* spp. [13] *Proteus* and *Enterobacter* spp. [14] are highly reported ESBL-producing bacteria worldwide. Nosocomial infections, often known as infections linked to hospitals or medical clinics, are thought to be the most common adverse event that endangers patient safety and has negative economic and societal repercussions. Nosocomial infection caused by *Pseudomonas* spp., *Klebsiella* spp., *Escherichia coli*, and *Staphylococcus* spp. [15]. 

The prevalence of beta-lactamases is being reported worldwide. β-lactams are commonly used broad-spectrum antibiotics with high efficacy, cost-effectiveness, easy delivery, and low adverse effects [16]. The risks of post-antimicrobial therapy have provoked policymakers to recognize the critical alert to human health and commit extra funding, progressively driving a resurgence of interest in antimicrobial discovery and improvement. The increased use of antibiotics can raise AMR against organisms, whereas multiple AMR has turned into a significant medical problem [17]. 

Additionally, hazardous metals from agrochemicals, industrial wastewater, and gas and coal mining industries can contaminate aquatic environments [18]. Because they accumulate via the food chain and pose risks to the environment, toxic metals are dangerous. The heavy metals are absorbed into enzymes and cofactors, making them necessary micronutrients for bacteria [1]. The aim of the current study was to see the effects of heavy metals on antibiotic susceptibility patterns of clinically isolated ESBL-producing bacteria of nosocomial origin.

## 2. Results

The sum of *n* = 300 clinical isolates was collected randomly from the microbiology laboratory of a tertiary care hospital in Lahore, Pakistan. These bacterial isolates were isolated from different types of clinical samples, as shown in Table 1. Most of the bacterial isolates were isolated from patients with urinary tract infections. The frequency of bacterial isolates is shown in Table 1. The type of specimen for these bacterial isolates and the demographic characteristics of infected patients has been shown in Table 1.

### The Phenotypic Confirmatory Test for the Synthesis of Extended-Spectrum Beta-Lactamase Was as Follows

The antibiotics susceptibility pattern of ESBL-producing bacteria has been seen in the disk diffusion test, as shown in Figure 1.

The test was considered ESBL positive when the bacteria were less sensitive to cefotaxime, and there was a clear increase in the inhibition zone of ceftriaxone in front of the Clavulanate-containing disc (Figure 2, left). This often creates a shape called a champagne cork or a keyhole, as shown in Figure 2 (right).

Most of the drugs were resistant, such as that enlisted by the CLSI-2020 for ESBL-producing bacteria, while, Colistin and Polymyxin-B were the only drugs that showed good efficacy against all isolated ESBL organisms. The direct antibiogram of tested bacterial isolates has been shown in Table 2.

ESBL-producing bacteria were analyzed as thick growth against Arsenic dilution 1.25 g/mL after incubation at 37 °C for 24 h. The heavy metals have significantly increased the resistance rate of antibiotics, as shown in Table 3.

## 3. Discussion

AMR is a major issue that might put the world in another pandemic. The misuse of antibiotics, excessive intake, and improper use are the major risk factors that contribute to the emergence of AMR [19]. The is a significant association between the misuse of antibiotics and the spread of AMR [20]. Bacteria’s genetic makeup may pass the AMR mechanisms within or between the bacterial family members or may also acquire via transportable genetic elements like plasmids from other spp. The AMR may spread between different bacterial strains as a result of horizontal gene transfer. Moreover, mutagenesis may also be responsible for causing AMR. Globally, antimicrobial drugs are widely prescribed drugs to treat nosocomial bacterial infections [21]. Apart from these factors, the presence of heavy metals in the environment may also play an important role in the emergence of MDR bacterial strains. Keeping in mind the scenario, the current study was conducted to see the prevalence of AMR strains responsible for infection of nosocomial origin in the heavy metal-containing bacterial growth agar medium. The AMR patterns of bacterial strains on with and without heavy metal treated agar were compared with each other.

The multiplication of ESBLs in recent years has significantly increased. The predominance of ESBL-producing *Klebsiella* spp. differs from country to country. In an overview of research facilities in the Netherlands under 1.5% of *E. coli* also, *K. pneumoniae* strains had an ESBL pattern [22]. While in France and Italy, ceftazidime obstruction was seen in as many as 45% of types of *K. pneumoniae* [9,23]. 

Antibiotic susceptibility and identification of the AMR agents implicated in the human body are critical for empirical treatment and the escaping of resistant bacteria [24]. Qamar et al. (2020) documented that antibiotics have enabled great advancements in healthcare systems, but they are under threat from rapidly evolving resistant microbes [25]. Only a few studies have been conducted in Pakistan to check the prevalence of metal-resistant bacteria and the antimicrobial sensitivity profiles of those bacteria [26,27,28]. The relationship between sensitivity and resistance patterns has also been studied previously [6]. One previous study used the double disc synergy test that verified the phenotypic resistance, which was the legitimate cause of the development of resistance to commercially available antibiotics [25]. The findings from previous studies indicated that heavy metal rates, which were elevated in the river tyne basin as a consequence of earlier industrial and mining activities, were related to the high rates of AMR [29,30].

The double-disc synergy test (DDST) was used in the current study to examine the isolates on Mueller-Hinton agar plates with 30 g/disc (containing 10 g of clavulanate) for possible patterns of the ESBLs. When compared to *Pseudomonas* spp., *Enterobacter* spp., *Proteus mirabilis*, and *Klebsiella* spp., the *E. coli* showed 20% isolates as ESBL-producing bacteria, while it was discovered that in total, around 50% of these bacteria produced the ESBLs. A study by Becerra-Castro et al. (2015) stated that the supplementation of metals in the culture medium reduced the culturability of *E. coli* by 95 and 98% [31]. Another study by Deredjianet al. (2011) showed that the strains showing strong resistance to antibiotics were the least resistant to metals [32]. However, the results of the current study showed that the metals significantly increased the AMR rates among tested bacteria. Heavy metals may enter the environment naturally or move there via anthropogenic contamination from both indirect and direct sources. Heavy metals may be released directly into the water, soil, and the environment through industrial activities, including mining, processing, and smelting.

A previous study from Canada elaborated that the health consequences of AMR include increased morbidity and mortality rates due to delays in starting effective treatments or treatment failure. People infected with antibacterial-resistant bacteria are two times more likely to be hospitalized and have long durations of hospital stay as compared to patients infected with susceptible strains [33]. Results of the current study showed that the tested bacterial isolates were 100% sensitive to certain antibiotics like colistin and polymyxin-B when tested without heavy metals. However, when tested on the heavy metals containing agar, these antibiotics also showed resistance. The possible key mechanisms for the possible link between the microbial acquisition of AMR and metal resistance are co-resistance and cross-resistance mechanisms.

The most commonly studied microorganisms with documented co-occurrences of antibiotic and heavy metal resistance are *Pseudomonas aeruginosa* and *E. coli* [34]. In contrast to water reservoirs, soils and sediments in various reservoirs have higher levels of heavy metals and antibiotic resistance. Abiotic variables like pH may also have an impact on the solubility of heavy metals for bacterial pathogens [5,34]. In the current study, when the organisms were tested for AST using heavy metal-containing agar, the *P. aeruginosa* isolates showed the highest resistance against tested antibiotics. *P. mirabilis* isolates were the second most common resistant strains. The possible factors that contribute to heavy metal pollution in environmental reservoirs include municipal wastewater, sludges, agricultural runoff, as well as industrial and pharmaceutical wastes. 

The co-existence of heavy metal and AMR genes among different bacteria makes the management of bacterial infections more challenging. Due to their usage in feed and as environmental contaminants, heavy metals are abundant using in livestock and livestock production systems, which has enabled many bacteria to acquire metal resistance [35]. A previous study by Pandit et al. (2020) has shown a significant correlation between high AMR rates and AMR genes [24]. Few studies have also reported that environmental factors could also be essential in helping bacteria to acquire AMR [9,36]. A study conducted by Abrar et al. (2019) has demonstrated that the virulent and AMR genes are usually associated with transposons or large plasmids. In addition, these plasmids usually carry AMR or other pathogenic factors such as toxins [19]. Verschuuren et al., (2021) have identified that the genes coding for AMR and enterotoxin were present on the same plasmid [22]. Hence, it is important to understand the relationship between heavy metals and AMR in various environmental reservoirs because environmental reservoirs are among the main channels by which antibacterial-resistant bacteria and antibacterial-resistant genes transmission to humans may occur and because of the complexity of AMR movement between and within these reservoirs.

## 4. Materials and Methods

The current study was conducted from January 2018 to July 2018 by the Department of Microbiology, University of Central Punjab, Lahore, Pakistan, under ID: L1F16MSMR0008. ESBL-producing samples were collected from a tertiary care hospital in Lahore. To re-identify and purify the collected bacterial spp., they were re-inoculated on selective media such as blood agar and MacConkey agar and incubated for 24–48 h at temperatures between 35 and 37 °C.

### 4.1. Collection of Bacterial Isolates

The bacterial isolates were collected from a tertiary care hospital in Lahore. The ESBL-producing organisms, including *Escherichia coli*, *Klebsiella* spp., *Pseudomonas aeruginosa*, *Proteus* spp., and *Enterobacter* spp., were collected, which were counter-identified later as ESBL-producing bacteria using the double disc synergy test.

### 4.2. Isolation and Re-Identification of Bacterial Isolates

All of the bacterial isolates were reinoculated on the cysteine electrolyte deficient (CLED) agar and MacConkey agar (Thermo Fisher Scientific, Inc., Waltham, MA, USA) and incubated at 37 °C for 18–24 h. After the incubation period, the bacterial colonies were evaluated for growth morphology and Gram staining characteristics. The final confirmation of bacterial isolates was done using biochemical tests-based identification. The biochemical tests, including citrate, indole, oxidase and analytical profile index 20E (API 20E) (BioMérieux, Marcy-l’Etoile, France), were used. The API 20E results were evaluated using the API website (https://apiweb.biomerieux.com/login) (accessed from 1 January 2018 to 31 December 2019).

### 4.3. Antibiotic Susceptibility Testing (AST) by the Kirby Bauer Disk Diffusion Method

The AST of bacterial isolated was done using Kirby Bauer disk diffusion methods as per the standard protocol from clinical laboratory standard institute (CLSI) guidelines 2020 [37]. Muller Hinton agar (MHA) (Thermo Fisher Scientific, Inc., Waltham, MA, USA) was used to perform the AST using a 0.5 MacFarland standard. The sterilized wire loop was used to pick the isolated 2–4 bacterial colonies from the culture plate. *E. coli*, *Proteus mirabilis*, *Klebsiella* Spp., and *Pseudomonas aeruginosa* single and identical colonies were chosen and moved into the MacFarland. A calibrated digital MacFarland meter was used to measure the turbidity of a microbiological cell in comparison to the supplied sample according to the standardized MacFarland method. After the preparation of MacFarland, it was lawned on the MHA plates, and then the antibiotic disks were dispensed on them. After the inoculation of plates and dispensing of antibiotics, these were incubated at 37 °C for 18–24 h. After the incubation period, the plates were checked for zones of inhibition (ZOIs). Results were noted as resistant (R), sensitive (S) and Intermediate (I).

The antibiotic disk (Thermo Fisher Scientific Inc., Waltham, MA, USA for ampicillin, amikacin, ceftriaxone, ceftriaxone, cefuroxime, chloramphenicol, ciprofloxacin, gentamicin, imipenem, meropenem, tetracycline, levofloxacin, tobramycin, fusidic acid, cefixime, colistin and polymyxin-B were tested.

### 4.4. The Double Disc Synergy Test (Phenotypic Confirmation Test)

ESBL-producing isolates were confirmed by the double disc synergy test (DDST). This was done in accordance with the instructions provided by the CLSI guidelines. The bacterial isolates were inoculated on MHA plates while simultaneously placing a disc of ceftriaxone (30 µg) and a disc of amoxicillin-clavulanate (10 µg) at a distance of 1 cm from each other. This allowed the test isolates to be exposed to both antibiotics at the same time (center to center). After overnight incubation at 37 °C, the plates were examined for phenotypic evidence of ESBL production. This was done by searching for an increase in the zone of inhibition of at least 5 mm between the cephalosporin discs and the amoxicillin-clavulanate discs corresponding to each of the cephalosporin discs.

### 4.5. Heavy Metals Susceptibility Pattern

Heavy metals (Arsenic) were purchased in the form of sodium arsenate (Disodium hydrogen arsenate heptahydrate) from Sigma-Aldrich, Massachusetts, United States. Generally, heavy metals are very toxic for living things, and these metals could be poisonous too. Heavy metals have the ability to degrade or inhibit the growth of certain microorganisms. Different materials were used to prepare a stock solution of arsenic salts, such as distilled water and flasks. The stock solution of arsenic (1.25 g/mL) was prepared as needed for the experimental procedures. The 10 g of sodium arsenate salt was added to 100 mL of autoclaved distilled water in a flask. This solution was mixed with 100 mL of MHA, and after this, the 20 mL of solution was poured onto Petri dishes. To check the AST of bacteria on heavy metal containing MHA plates at a concentration of 1.25 g/mL, the same procedure was repeated as mentioned above. After 18–24 h of incubation period at 37 °C, the ZOIs were measured to determine the antibiotic susceptibility pattern [38].

### 4.6. Statistical Analysis

The data was entered in SPSS version 26.0 (IBM, New York, NY, USA). At first, the descriptive analysis was applied to check the frequency (*n*), percentage (%), mean, and standard deviation (SD). The chi-square test was run to see the difference among the studied variable. A *p*-value of <0.05 was considered statistically significant.

## 5. Conclusions

AMR is a worldwide health-related issue these days. The effect of antibiotics becomes lesser due to resistant mechanisms developed by bacteria. The over-administration, misuse and wrongly prescribed antibiotics lead to a worsening situation for human beings. It was found in the current study that a significant frequency of ESBL-producing bacteria was discovered in clinical isolates, and these bacteria had a high ratio of resistance to tested antibiotics. MDR-ESBL has created a great threat under the edge of the high AMR rates. Furthermore, the excessive rate of heavy metal-induced AMR has increased the risk of getting worse the situation of AMR. Current findings confirm that heavy metals contribute significantly to the rise in AMR rate. These heavy metals may also be present in the environment also, which may pose a serious risk of higher AMR rates. To further understand the exposure-response linkages between heavy metals and AMR in various environmental media, more research studies using statistical data are required. It is recommended that culture-based and molecular-based approaches be used together in future research to learn more about how bacteria can be resistant to both heavy metals and antibiotics. 

## Figures and Tables

**Figure 1 pharmaceuticals-15-01426-f001:**
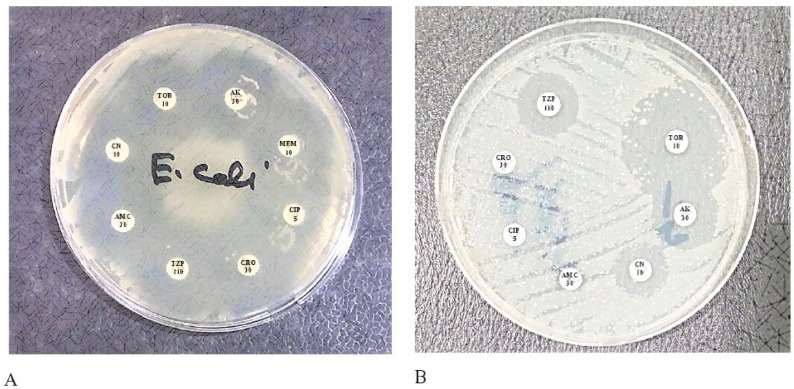
The transformation between ESBL and Non-ESBL bacteria. Ceftriaxone (CRO) antibiotic is a third-generation cephalosporin that is reactive to non-ESBL strains but Non-reactive to ESBL strains; (**A**) is a non-ESBL strain so susceptible to CRO; (**B**) is an ESBL strain and CRO resistant.

**Figure 2 pharmaceuticals-15-01426-f002:**
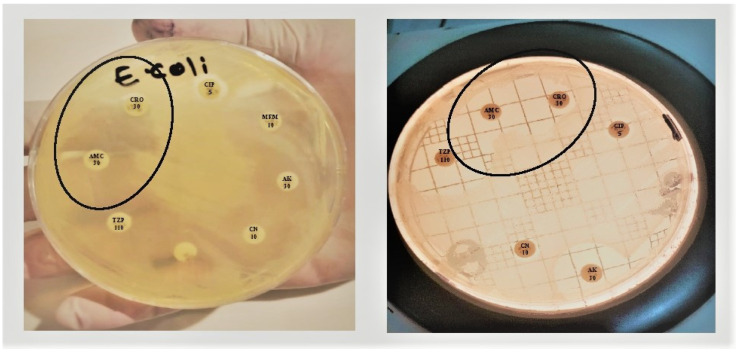
Double disc synergy test (DDST): A decreased susceptibility to ceftriaxone (CRO) is combined with a clear-cut enhancement of the ZOI in front of the Amoxicillin Clavulanate (AMC) containing disk as showed in the circle, often resulting in a characteristic shape-zone referred to as ‘champagne-cork’ or ‘keyhole.’.

**Table 1 pharmaceuticals-15-01426-t001:** Distribution of study variables.

Characteristics		Frequency (*n*)	Percentage (%)	*p*-Value
Type of specimens	Wound	63	21	0.002
	Urine	112	37.3	
	Blood	43	14.3	
	Pus	49	16.3	
	Aspirate	33	11	
Clinical isolate	*Enterobacter* spp.	18	6	0.008
	*Proteus mirabilis*	21	7	
	*P. aeruginosa*	32	10.6	
	*Klebsiella* spp.	57	19	
	*E. coli*	172	57.3	
Age (Years)	18–23	63	21	0.002
	23–28	112	37.3	
	28–33	43	14.3	
	33–38	49	16.3	
Gender	Male	189	63	0.003
	Female	111	37	
Enrollment	Indoor Patients	204	68	
	Outdoor Patients	96	32	

**Table 2 pharmaceuticals-15-01426-t002:** Percentage resistance profile of clinically isolated bacteria.

Antibiogram	Disc Contents	*E. coli* (*n* = 172)	*Klebsiella* Spp. (*n* = 57)	*P. aeruginosa* (*n* = 32)	*Enterobacter* Spp. (*n* = 18)	*Proteus mirabilis* (*n* = 21)
Ampicillin	10 µg	93.1	96.3	91	87	92
Amikacin	30 µg	82	87.3	79.5	87	74
Ceftriaxone	30 µg	75	82	68	77	96
Cefuroxime	30 µg	97	92.4	92	90	97
Chloramphenicol	30 µg	99	97	77	98	92
Ciprofloxacin	5 µg	94	96	93.5	89.2	98
Gentamicin	10 µg	93.5	88	91.2	83.5	99
Imipenem	10 µg	91.5	92.2	97	99	81.5
Meropenem	10 µg	90	98	99	90.5	77
Tetracycline	30 µg	92.7	91.3	92	89.2	99
Levofloxacin	50 µg	89.5	97	96	99	92.5
Tobramycin	10 µg	92	98	98.5	93.5	98
Fusidic acid	10 µg	91	88.5	99	90	97
Cefixime	5 µg	88	99	98	99	87.5
Colistin	0	0	0	0	0	0
Polymyxin B	0	0	0	0	0	0

**Table 3 pharmaceuticals-15-01426-t003:** Heavy metals (Arsenic) induced resistance against the CLSI-approved ESBL antibiogram panel.

Antibiogram	Heavy Metals Concentration	*E. coli* (*n* = 172)	*Klebsiella* Spp. (*n* = 57)	*P. aeruginosa* (*n* = 32)	*Enterobacter* Spp. (*n* = 18)	*Proteus mirabilis* (*n* = 21)
Ampicillin	1.25 g/mL	100	100	100	96	100
Amikacin		100	100	100	100	88
Ceftriaxone		100	78	89	84	100
Cefuroxime		92	100	100	100	100
Chloramphenicol		100	100	73	100	100
Ciprofloxacin		89	100	100	100	100
Gentamicin		100	97	100	97	100
Imipenem		86	100	100	100	98
Meropenem		100	83	100	100	83
Tetracycline		83	100	93	96	100
Levofloxacin		100	97	100	97	100
Tobramycin		93	91	100	100	100
Fusidic acid		100	100	100	100	100
Cefixime		100	100	100	100	100
Colistin		32	23	56	33	12
Polymyxin B		47	18	42	13	16

## Data Availability

The data relating to the current study can be accessed upon reasonable request to the corresponding author.

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
