# Peer review of "Heavy Metal (Arsenic) Induced Antibiotic Resistance among Extended-Spectrum β-Lactamase (ESBL) Producing Bacteria of Nosocomial Origin"

_pharmaceuticals, 2022, doi:10.3390/ph15111426_

Round 1
Reviewer 1 Report
A manuscript with the title "Heavy metal (Arsenic) induced antibiogram profiles in ex- 2 tended-spectrum β-lactamase (ESBL) producing bacteria" has shown the role of heavy metal (arsenic) to be one of the primary causes of antimicrobial resistance (AMR) along with the excessive and improper use of antibiotics. The co-contamination of heavy metals and antibiotic compounds existing in the environment might also be exponentiating the spread of AMR.
This manuscript has basic issues to be resolved by the researchers.
1. Bacterial strains have been reisolated, but have not been identified at the molecular level using 16S-rRNA sequencing. It is crucial to identify the strains by gene sequencing., to be sure.
2. Methodology paper has not been written properly. re-isolation, purification, and biochemical characterization have all been mixed up and confused. All of the steps should have been rewritten properly in detail.
3. Heavy metal salt name is missing, the stock solution concentration is missing, and the concentration used in the media plates is missing.
4. Company source is missing fo chemicals and a heavy metal salt. Thus experiments can not be reproducible.
5. Moreover, the phenotypic presentation of all other isolates is missing.
6. english has to be improved.
7. conclusion has to be focused and elaborated and linked to the outcomes of the findings to the future prospects and recommendations.
Author Response
Reviewer 1
Comments and Suggestions for Authors
A manuscript with the title "Heavy metal (Arsenic) induced antibiogram profiles in ex- 2 tended-spectrum β-lactamase (ESBL) producing bacteria" has shown the role of heavy metal (arsenic) to be one of the primary causes of antimicrobial resistance (AMR) along with the excessive and improper use of antibiotics. The co-contamination of heavy metals and antibiotic compounds existing in the environment might also be exponentiating the spread of AMR. This manuscript has basic issues to be resolved by the researchers.
- Bacterial strains have been reisolated, but have not been identified at the molecular level using 16S-rRNA sequencing. It is crucial to identify the strains by gene sequencing., to be sure.
Response: Dear reviewer, as we discussed in the manuscript in the lines number 257-268 in the methodology section. We confirmed ESBL-producing organisms based on the double disc synergy test as recommended by CLSI in 2020 guidelines under the definition “a disc containing 30 µg/disc of ceftriaxone and a disc containing amoxicillin–Clavulanate (10 µg of Clavulanate) at a distance of 1 cm from the test isolates. This allowed the test isolates to be exposed to both antibiotics at the same time (centre to centre). After an overnight incubation at 37°C, the plates were examined for phenotypic evidence of ESBL production. This was done by searching for an increase in the zone of inhibition of at least five millimetres between the cephalosporin discs and the amoxicillin-Clavulanate discs corresponding to each of the cephalosporin discs. This was done in order to determine whether or not ESBL was being produced” so after that statement and methodology of CLSI-2020, it was not necessary to perform the molecular identification due to chances of contamination, time consuming, environmental condition and factors that’s why we followed for phenotypic confirmatory test as per standard of CLSI-2020 guidelines.
For the basic confirmation of bacterial isolates, standard methods like colony morphology, biochemical testing including API testing and Gram staining were used. And then the ESBL confirmation was confirmed as mentioned above.
Furthermore, we agree that the 16SRNA sequencing could be the best tool for the identification of isolates, but unfortunately because of financial limitations and the less availability of material we were unable to perform the sequencing. We focused on the gold standard microbiological identification instead.
- Methodology paper has not been written properly. re-isolation, purification, and biochemical characterization have all been mixed up and confused. All of the steps should have been rewritten properly in detail.
Response: (Line 229-248, 254-267, 274-290) Dear reviewer, thank you for highlighting the valuable points and suggestions to improve the manuscript. The methodology section has been revised significantly and every protocol has been elaborated in the revised version of manuscript. The section about reinoculation and reidentification of bacterial isolated has been revised.
- Heavy metal salt name is missing, the stock solution concentration is missing, and the concentration used in the media plates is missing.
Response: Line 274-285: The metal salt name has been added in the revised version of manuscript. Furthermore, the stock solution concentration, and the concentration used in the media plates has been mentioned.
- Company source is missing for chemicals and a heavy metal salt. Thus experiments cannot be reproducible.
Response: Line 274: The company name of the salt has been mentioned in the revised version of manuscript.
- Moreover, the phenotypic presentation of all other isolates is missing.
Response: Line 244-266: The section about phenotypic presentation of bacterial isolates for AMR patterns and ESBLs identification has been added in the revised version of manuscript.
- English has to be improved.
Response: Dear reviewer, thank you for highlighting the valuable issue about the requirements of English proofreading. The revised version of manuscript has been thoroughly revised for English proofreading and grammatical mistakes.
- Conclusion has to be focused and elaborated and linked to the outcomes of the findings to the future prospects and recommendations.
Response: Line 292-295, 300-307: The conclusion section has been thoroughly revised in the revised version of manuscript.

Reviewer 2 Report
The manuscript entitled ‘Heavy metal (Arsenic) induced antibiogram profiles in extended-spectrum β-lactamase (ESBL) producing bacteria’ provides interesting area of research. The manuscript may be modified with the help of an English-speaking native researcher.
Title: The authors have attempted the antibiogram profile of the bacterial isolates of nosocomial origin; hence a modification in the title would highlight the impact of the study.
Abstract: The grammatical errors and typos may be looked into.
Introduction:
- Line 60: Cite proper abbreviation for antibiotic resistance. The abbreviation once cited can be used hereafter in the manuscript.
- Several typos and grammatical errors appers in the text.
- The mention/ importance of nosocomial pathogens could be explained in the introduction section to make the readers more interested to the topic.
Results:
- The antibiogram profile of the individual isolate, if supplied as supplementary table, would improve the expression of results.
Discussion:
- Overall, the discussion is poorly attempted. There is no mention of the heavy-metal induced drug resistance. Albeit the authors cited published literature, the theoretical/mechanistic background for the same is missing.
- Lines 142-143: The authors have stated many previous studies…; however, a single study has been cited.
- Line 144: ‘close family members’ what does it imply?
Methods:
- Several typos are seen in the methodology section (eg: line 237: MacFarlane instead of MacFarland).
Author Response
Reviewer 2
Comments and Suggestions for Authors
The manuscript entitled ‘Heavy metal (Arsenic) induced antibiogram profiles in extended-spectrum β-lactamase (ESBL) producing bacteria’ provides interesting area of research. The manuscript may be modified with the help of an English-speaking native researcher.
Response: Dear reviewer, thank you for highlighting the valuable issue about the requirements of English proofreading. The revised version of manuscript has been thoroughly revised for English proofreading and grammatical mistakes.
Title: The authors have attempted the antibiogram profile of the bacterial isolates of nosocomial origin; hence a modification in the title would highlight the impact of the study.
Response: Dear reviewer, thank you for you suggestion to make the title more attractive. The title has been corrected as per the suggestions.
Abstract: The grammatical errors and typos may be looked into.
Response: The abstract has been revised thoroughly and carefully checked for English proofreading and grammatical mistakes.
Introduction:
- Line 60: Cite proper abbreviation for antibiotic resistance. The abbreviation once cited can be used hereafter in the manuscript.
- Several typos and grammatical errors appears in the text.
- The mention/ importance of nosocomial pathogens could be explained in the introduction section to make the readers more interested to the topic.
Response: The manuscript has been revised for abbreviation at their first appearance. It was carefully checked in the revised version of manuscript. For antimicrobial resistance (AMR) the abbreviation has been mentioned at Line 39, and from Line 39 to onward only AMR was used in the revised version of manuscript.
The manuscript has been thoroughly revised for English proofreading and grammatical mistakes.
Line 87-90: An explanation about nosocomial infections has been added in the revised version of manuscript.
Results:
- The antibiogram profile of the individual isolate, if supplied as supplementary table, would improve the expression of results.
Response: Dear reviewer, the antibiotic susceptibility patterns of the current tested bacterial isolates has been elaborated in Table 2. Furthermore, we apologies that the full year institutional antibiogram we can not provide here because of ethical consideration from the institution.
Discussion:
- Overall, the discussion is poorly attempted. There is no mention of the heavy-metal induced drug resistance. Albeit the authors cited published literature, the theoretical/mechanistic background for the same is missing.
- Lines 142-143: The authors have stated many previous studies…; however, a single study has been cited.
- Line 144: ‘close family members’ what does it imply?
Response: Line 153-166, 175-178, 179, 181-213: The discussion section has been revised thoroughly. Furthermore, it has been revised for English proofreading. More studies have been added in the discussion section to discuss with the results of current study.
Methods:
- Several typos are seen in the methodology section (eg: line 237: MacFarlane instead of MacFarland).
Response: Dear reviewer, thank you for highlighting the typing error throughout the manuscript. The manuscript has been thoroughly revised for grammatical mistakes.

Round 2
Reviewer 2 Report
Although the authors have considerably improved the manuscript, some of the errors are lying pending and is pointed out for modification:
1. Line 64: AMR; Line 82: ESBLs- Use the expanded form while using in the manuscript for the first time.
2. Line 91. It shall be Staphylococcus; follow the scientific nomenclature.
3. Line 92: Use beta-
4. Line 97: The increased use of antibiotics can raise.....
5. Line 104: The aim shall be modified including the nosocomial infections.
6. Line 184: was used...
7. Although the discussion has remarkably been improved, the possibility of heavy metal- induced drug resistance is still lacking in the light of observations made in the present study. It would be better to discuss the same in connection with the hypothesized drivers of AMR.
Author Response
Reviewer 2
Comments and Suggestions for Authors
Although the authors have considerably improved the manuscript, some of the errors are lying pending and is pointed out for modification:
- Line 64: AMR; Line 82: ESBLs- Use the expanded form while using in the manuscript for the first time.
Response: Line 64 and 82: The abbreviations have been mentioned.
L2. Line 91. It shall be Staphylococcus; follow the scientific nomenclature.
Response: Line 91: staphylococcus has been corrected as “Staphylococcus” and species have been replaced with “spp.” Throughout the manuscript.
- Line 92: Use beta-
Response: Line 92: Corrected as “beta”
- Line 97: The increased use of antibiotics can raise.....
Response: Line 97: The sentence has been corrected.
- Line 104: The aim shall be modified including the nosocomial infections.
Response: Line 105: The aim has been modified as suggested.
- Line 184: was used...
Response: Line 184: “are” is replaced with “were”
- Although the discussion has remarkably been improved, the possibility of heavy metal- induced drug resistance is still lacking in the light of observations made in the present study. It would be better to discuss the same in connection with the hypothesized drivers of AMR.
Response: Line 185-189, 197-199, 207-209 and 221-225. Discussion section has been improved by adding more discussion points.